# A deep generative model of 3D single-cell organization

Rory M. Donovan-Maiye[1¤a], Jackson M. Brown[1], Caleb K. Chan[1], Liya Ding[1¤b], Calysta Yan[1], Nathalie Gaudreault[1], Julie A. Theriot[1,2], Mary M. Maleckar[1¤c], Theo A. Knijnenburg[1]*, Gregory R. Johnson[1¤d]

1 Allen Institute for Cell Science, Seattle, Washington, United States of America, 2 Department of Biology and Howard Hughes Medical Institute, University of Washington, Seattle, Washington, United States of America

¤a Current address: Novo Nordisk, Seattle, Washington, United States of America
¤b Current address: Southeast University, Nanjing, China
¤c Current address: Simula Research Laboratory, Oslo, Norway
¤d Current address: Amazon Web Services, Seattle, Washington, United States of America
* theo.knijnenburg@alleninstitute.org

**Data Availability Statement:** All raw data files are available from https://open.quiltdata.com/b/allencell/packages/aics/pipeline_integrated_single_cell and https://www.allencell.org/. All code files are

## Abstract

We introduce a framework for end-to-end integrative modeling of 3D single-cell multi-channel fluorescent image data of diverse subcellular structures. We employ stacked conditional $\beta$-variational autoencoders to first learn a latent representation of cell morphology, and then learn a latent representation of subcellular structure localization which is conditioned on the learned cell morphology. Our model is flexible and can be trained on images of arbitrary subcellular structures and at varying degrees of sparsity and reconstruction fidelity. We train our full model on 3D cell image data and explore design trade-offs in the 2D setting. Once trained, our model can be used to predict plausible locations of structures in cells where these structures were not imaged. The trained model can also be used to quantify the variation in the location of subcellular structures by generating plausible instantiations of each structure in arbitrary cell geometries. We apply our trained model to a small drug perturbation screen to demonstrate its applicability to new data. We show how the latent representations of drugged cells differ from unperturbed cells as expected by on-target effects of the drugs.

## Author summary

It is impossible to acquire all the information we want about every cell we are interested in in a single experiment. Even just limiting ourselves to imaging, we can only image a small set of subcellular structures in each cell. If we are interested in integrating those images into a holistic picture of cellular organization *directly from data*, there are a number of approaches one might take. Here, we leverage the fact that of the three channels we image in each cell, two stay the same across the data set; these two channels assess the cell's shape and nuclear morphology. Given these two reference channels, we learn a model of cell and nuclear morphology, and then use this as a reference frame in which to learn a representation of the localization of each subcellular structure as measured by the third

available from GitHub at https://github.com/AllenCellModeling/pytorch_integrated_cell.

**Funding:** The study was supported by the Allen Institute. The funders had no role in study design, data collection and analysis, decision to publish, or preparation of the manuscript.

**Competing interests:** The authors have declared that no competing interests exist.

channel. We use $\beta$-variational autoencoders to learn representations of both the reference channels and representations of each subcellular structure (conditioned on the reference channels of the cell in which it was imaged). Since these models are both probabilistic and generative, we can use them to understand the variation in the data from which they were trained, to generate instantiations of new cell morphologies, and to generate imputations of structures in real cell images to create an integrated model of subcellular organization.

## 1 Introduction

Decades of biological experimentation, coupled with ever-improving advances in microscopy, have led to the identification and description of many subcellular structures that are key to cellular function. Understanding the unified role of these component structures in the context of the living cell is indeed a goal of modern-day cell biology. How do the multitude of heterogeneous subcellular structures localize in the cell, and how does this change during dynamic processes, such as the cell cycle, cell differentiation and the response to internal or environmental perturbations [1, 2]? A comprehensive understanding of global cellular organization remains challenging, and no unified model currently exists.

Advances in microscopy and live cell fluorescence imaging in particular have led to insight and rich data sets with which to explore subcellular organization. However, the experimental state-of-the-art for live cell imaging is currently limited to the simultaneous visualization of only a limited number (2–6) of tagged molecules. Additionally, there are substantial, interdependent limitations regarding spatial and temporal resolution as well as duration of live cell imaging experiments. Computational approaches offer a powerful opportunity to mitigate these limitations by integrating data from diverse imaging experiments into a single model, a step toward an integrated representation of the living cell and additional insight into its function.

Computational models in this domain can be divided into those that operate directly on the microscopy images of cells, and those that are based on descriptors of texture or segmented objects extracted from the image data. The image feature-based methods of the latter category have previously been employed to describe and model cellular organization [3–5]. These approaches use accurate object segmentations to convey detailed information about the size, shape and localization of subcellular structures. Importantly, segmentation procedures must be judiciously designed for each type of structure and significant amounts of effort may be spent designing features to be useful for a specific task (e.g. to measure "roughness" of a structure). Ground truth data for evaluation of segmentation and feature selection can be difficult to obtain, especially for 3D cell images [6]. These challenges compound when trying to expand models to describe relationships between multiple subcellular structures and their organization within a cell.

Recent deep-learning approaches that operate directly on the image data have become increasingly popular in multiple cell biology applications and provide an alternate computational pathway towards integrated visual representations of the cell. In cell imaging, deep neural networks (DNNs) can be utilized to perform pixel-level tasks, such as object segmentation [7], label-free prediction in 2D and 3D images [8, 9], de-noising and image restoration [10, 11], or cell-level analyses such as predicting cell fates [12], classifying cell cycle status [13], distinguishing motility behaviors of different cell types [14], and subcellular pattern representation [15]. It should be noted that *generative models* of individual cells are particularly useful for building an integrated representation of the living cell, as these models can capture how

relationships among subcellular structures vary across a population of cells and encode these as distributions. Generative models based on image features have been used to understand the spatial distribution of organelles under different conditions [1, 16–18]. Deep generative models that operate directly on the image data have been growing in popularity in the 2D image domain and have been shown to be useful for image synthesis [19], lineage mapping [20], predicting morphological effects of drugs [21] and multi-modal data harmonization [22].

Our work expands upon these efforts; by combining the results of microscopy experiments measuring the localization of independent subcellular structures in 3D images, we produce a deep generative model of 3D single-cell organelle localization conditioned on cell and nuclear morphology. While generative models using segmentation-based image features allow us to explore the feature and shape space of the cells, using the actual images directly, i.e. the fluorescent intensity values, allows us to include all visible features that are captured through microscopy and directly synthesize cell images with microscopy-level details. In addition, using 3D microscopy images as our input allows us to model the cells and intracellular structures directly in 3D space without having to limit them to 2D projections. Therefore, our model is useful for both image synthesis and interpretable representations and therefore may be utilized for several downstream tasks, including visualization, dimensionality reduction, quantifying changes in cell organization as a function of cell state (mitotic state, drug treatment, etc.), and the estimation of statistical relationships between structures observed simultaneously. Notable in our approach is that we can build our model without the necessity of laborious hand-crafted features or specialized subcellular segmentations.

Below, we explain how the Statistical Cell is constructed; we discuss its useful, novel contributions and provide a critical look at its current limitations.

## 2 Results

### 2.1 Statistical cell: A variational autoencoder that models the 3D organization of subcellular structures

In this section we begin with an overview of the model, and then proceed to present its ability to model cell morphology as well as the localization of subcellular structures.

In order to jointly model the variation of all subcellular structures in our data, we engineered a stacked conditional $\beta$-variational autoencoder and trained it end-to-end on the entirety of our data. We call this model the Statistical Cell. The Statistical Cell is a data-driven probabilistic model of the organization of the human cell membrane, nuclear shape and subcellular structure localization. The diverse array of subcellular structures used here represent components that serve specific functions that may be useful for understanding cellular state.

The organizing principle of our model is that the localization of subcellular structures is meaningful only in relation to the cell geometry in which they are embedded. By leveraging conditional relationships to the cell and nucleus, we allow for the integration of different subcellular structures into a single model, without these structures needing to be tagged and imaged simultaneously.

The model is trained on a collection of more than 40,000 high-resolution 3D images of live human induced pluripotent stem cells (Fig 1A). Using 3D spinning disk confocal microscopy we collected three image channels for each of these cells: 1) Plasma membrane using CellMask Deep Red dye, 2) Nucleus via labeling DNA with NucBlue Live dye, and 3) one of 24 subcellular structures. Specifically, these cells are from clonal lines, each gene-edited to endogenously express a fluorescently tagged protein that localizes to a specific subcellular structure. To facilitate biological interpretation of our model, 5 of the 24 structures are synthetic controls, where instead of a normal GFP structure channel we present to the model either a blank channel as a

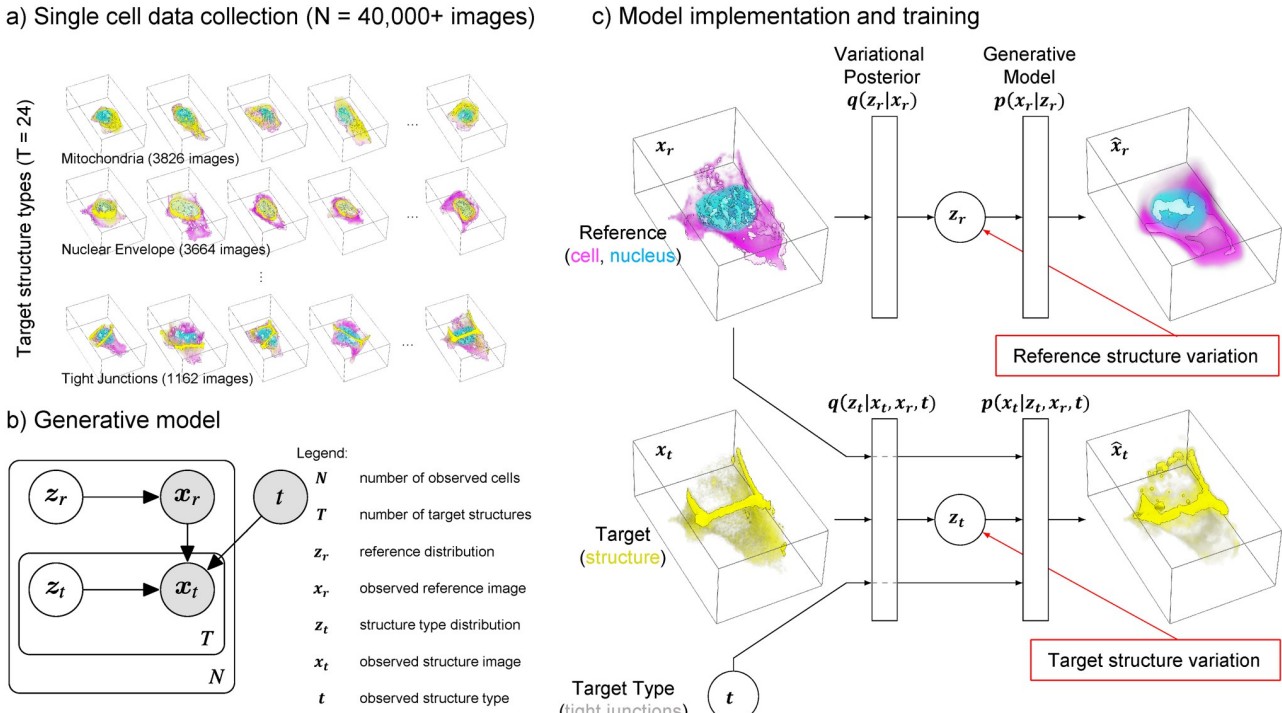

**Fig 1. The statistical cell.** a) A visual overview of the single-cell data collection used in this study. For each of more than 40,000 cells we have high-resolution 3D image data of the shape and location of the cell membrane (pink), nucleus (blue) and one of 24 endogenously tagged subcellular structures (yellow). The examples show actual image data of cells in the collection. b) Plate diagram of the Statistical Cell model. Shaded and un-shaded circles represent observed and learned variables, respectively. We model reference structures $x_r$ as draws from a latent variable $z_r$, and target structures $x_t$ as draws from the latent variable $z_t$, conditioned on $x_r$ and target type $t$. c) The main components of the models are two autoencoders: one encoding the variation in the reference, i.e. the cell and nuclear shape (top), and another which learns the relationship between the target (the subcellular structures) dependent on the reference encoding (bottom).

structure, duplicated membrane and DNA channels, random Gaussian noise in the membrane and DNA regions, or copy a random structure channel from a different cell. There are between 1,000–4,000 cells per structure type (Table 1).

Using these images, we model cell membrane, DNA, and subcellular structure given a known target structure type (i.e. a model of cell shape, nuclear shape, and structure organization, given that the structure is one of the 24 observed structure types). This model takes the form $p(x_{r,t}|t)$ where $x$ is an image, $r$ indicates *reference*, i.e. the reference cell membrane and DNA dyes, $t$ indicates the *target* channel. $x_{r,t}$ is therefore an image of a cell containing reference structures (membrane, DNA) and a target structure (one of the 24 structure types).

Utilizing relationships in our data (Fig 1A and 1B), the model is factored into two subcomponents; a reference model $\mathbf{M_R}$ that maximizes the probability of observed cell and DNA organization $p(x_r)$, and a target model $\mathbf{M_T}$ that maximizes the conditional probability of subcellular structure organization $p(x_t|x_r, t)$. The complete probability model is therefore $p(x_{r,t}|t) = p(x_r)p(x_t|x_r, t)$.

Each component is modeled with a variational autoencoder (Fig 1C) [23], allowing us to generate integrated examples from the learned data distribution, as well as map reference $x_r$ and target $x_t$ to learned low dimensional variables (or embeddings / latent space / latent dimensions) $z_r$ and $z_t$, that capture morphological variation and relationships between the reference and target structures. It is important to note that $z_t$ is a *conditional* embedding, i.e. a

**Table 1. Structures used in this study, with corresponding genes, number of images and labels.** Each image contains the labeled structure in addition to channels of labeled cell and nuclear shape.

| Structure Name | Gene Name (gene symbol) | # Images | Labels | |
|---|---|---|---|---|
| | | | #Interphase | #Mitosis |
| Actin filaments | actin beta (ACTB) | 2,848 | 2,544 | 304 |
| Actomyosin bundles | myosin heavy chain 10 (MYH10) | 1,392 | 1,282 | 110 |
| Adherens junctions | catenin beta 1 (CTNNB1) | 2,343 | 2,202 | 141 |
| Centrioles | centrin 2 (CETN2) | 1,605 | 1,405 | 200 |
| Desmosomes | desmoplakin (DSP) | 2,320 | 2,161 | 159 |
| Endoplasmic reticulum | SEC61 translocon beta subunit (SEC61B) | 1,120 | 1,045 | 75 |
| Endosomes | RAB5A, member RAS oncogene family (RAB5A) | 1,562 | 1,455 | 107 |
| Gap junctions | gap junction protein alpha 1 (GJA1) | 1,491 | 1,334 | 157 |
| Golgi | ST6 beta-galactoside alpha-2,6-sialyltransferase 1 (ST6GAL1) | 1,539 | 1,445 | 94 |
| Lysosomes | lysosomal associated membrane protein 1 (LAMP1) | 1,476 | 1,309 | 167 |
| Matrix adhesions | paxillin (PXN) | 1,637 | 1,531 | 106 |
| Microtubules | tubulin-alpha 1b (TUBA1B) | 2,409 | 2,219 | 190 |
| Mitochondria | translocase of outer mitochondrial membrane 20(TOMM20) | 3,826 | 3,590 | 236 |
| Nuclear envelope | lamin B1 (LMNB1) | 3,664 | 3,455 | 209 |
| Nucleoli Dense Fibrillar Component (DFC) | fibrillarin (FBL) | 1,536 | 1,407 | 129 |
| Nucleoli Granular Component (GC) | nucleophosmin 1 (NPM1) | 3,717 | 3,480 | 237 |
| Peroxisomes | solute carrier family 25 member 17 (SLC25A17) | 1,455 | 1,369 | 86 |
| Plasma membrane | Safe harbor locus (AAVS1) (CAAX domain of K-Ras) | 2,098 | 1,867 | 231 |
| Tight junctions | tight junction protein 1 (TJP1) | 1,162 | 1,079 | 83 |
| Control—Blank | N/A | 2,028 | 1,861 | 167 |
| Control—DNA | N/A | 2,028 | 1,861 | 167 |
| Control—Memb | N/A | 2,028 | 1,861 | 167 |
| Control—Noise | N/A | 2,028 | 1,861 | 167 |
| Control—Random | N/A | 2,028 | 1,861 | 167 |
| Total | | 49,340 | 45,484 | 3,856 |

low dimensional representation of information in the target image, given the information in the reference and the specific target type. For details see Section 4.2.

## 2.2 Representing and visualization of subcellular organization via latent space embeddings

The $\beta$ variational autoencoder ($\beta$-VAE) architecture underlying the model of cell and nuclear shape ($\mathbf{M_R}$) compresses the variation in the 3D cell images into a subspace with maximal dimensionality of 512. While each cell is described by 512 coefficients in this latent space, the diagonal covariance matrix of the Gaussian prior of the latent embeddings, in tandem with the KL-divergence term in the $\beta$-VAE objective function (Eq 3) attempts to sparsify the latent space by penalizing spurious embedding dimensions. In general we find that fewer than 100 dimensions (out of 512) show non-zero KL-divergence terms, effectively reducing the dimensionality from hundreds of thousands of voxels in the original images to fewer than 100 latent space coefficients (S6 and S3 Figs). Different values of $\beta$ lead to different numbers of effective latent dimensions; see S6 Fig for details. (We note there is also a latent space for $\mathbf{M_T}$, but we limit our analysis and discussion in this section to the latent space of the reference model $\mathbf{M_R}$).

We investigated how the latent space of the model represents cell morphology by correlating the cell coefficients for each latent dimension with measured cell metrics, such as cell size and mitotic state. We observed various strong correlations between cell metrics and those latent dimensions that showed a significant variation; see S3 Fig for details. As shown in Fig 2, the two top dimensions of the latent space visually stratify the cells based on mitotic stages as well as cell height. Specifically, cells late in prometaphase and metaphase are generally taller and rounder than the bulk of interphase cells, while newborn daughter cells (annotated as anaphase / telophase / cytokinesis) typically have smaller nuclei than most other interphase cells, which may be in the middle of DNA replication. Importantly, cell images that are generated

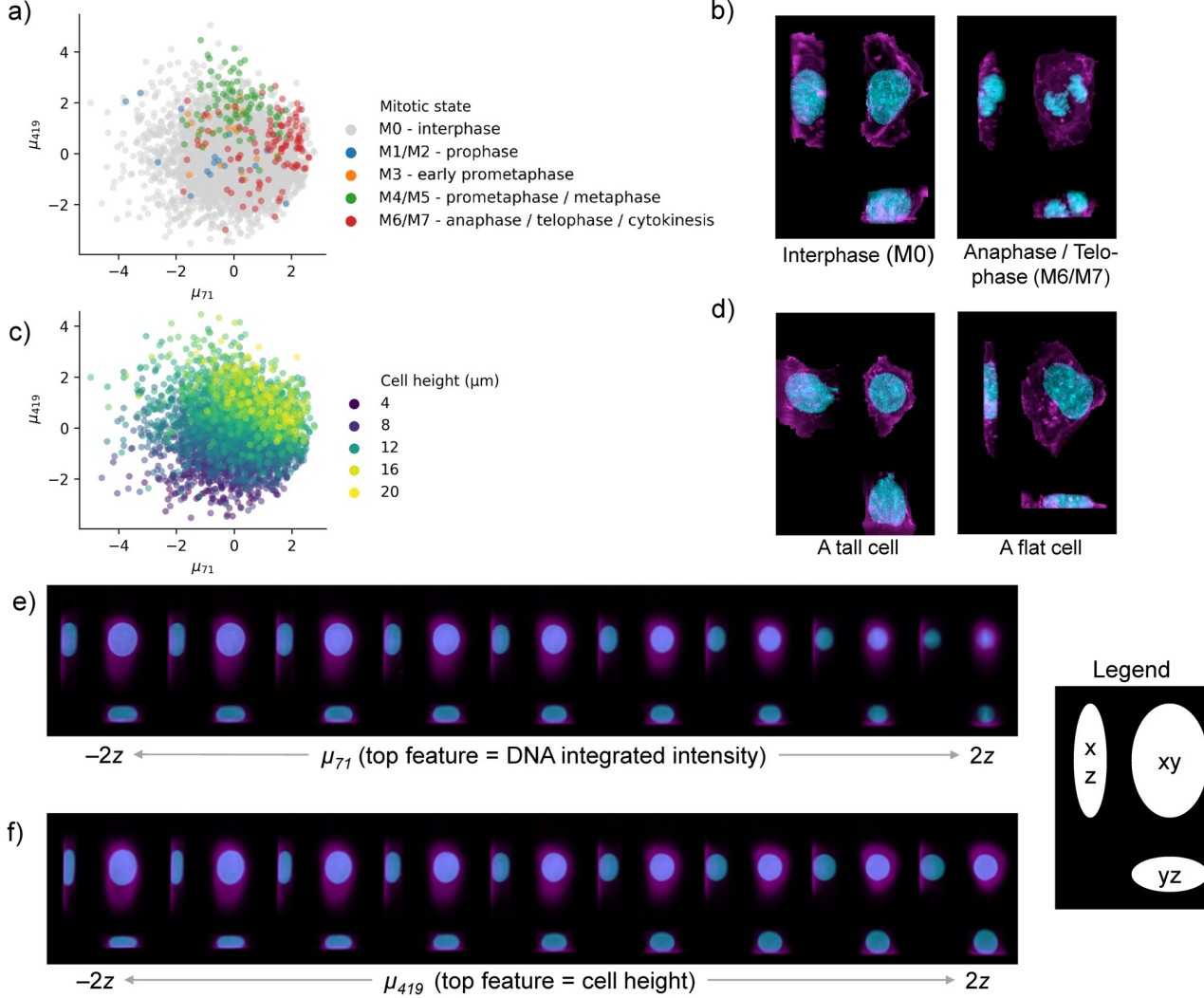

**Fig 2. The reference latent space learned by the model encodes interpretable features and stratifies cells by biologically relevant features.** a) Cells in the test set undergoing mitosis are stratified by the top two dimensions of the reference model latent space. b) Maximum intensity projections of two reals cell; one in interphase (left) and one undergoing cell division (right). c) The top two dimensions of the reference latent space (same as in a)) for all cells in the test set, colored by cell height. d) Maximum intensity projections of two reals cell; a tall cell (left) and flat cell (right). e) Maximum intensity projections of 9 generated 3d cell images along latent space dimension $\mu_{71}$. The integrated intensity of the DNA channel correlates strongly with $\mu_{71}$. f) Similar to e) but now showing latent space dimension $\mu_{419}$ which correlated strongly with cell height. The latent walks in parts e) and f) occur in nine steps that span -2 to 2 deviations of that latent dimension's variation. See S1 and S2 Figs for more visual associations between latent dimensions and features, and S3 Fig for exhaustive correlations of features with latent dimensions.

along these latent space dimensions show the expected phenotype (Fig 2E and 2F). It is worthwhile to note that the model captures both biologically interpretable features, such as cell height, as well as features that are mainly properties of the image measurement such as the overall fluorescent dye intensity. Specifically, the DNA integrated intensity (which correlates with $\mu_{71}$ as visualized in Fig 2E) is the total brightness of the DNA channel for a particular cell. There are at least three considerations that may lead to variation in total DNA integrated intensity for our data set: 1) labeling of the DNA with Hoechst dye may vary from cell to cell and from one imaging session to another, 2) as cells progress through S phase (DNA replication) the total amount of DNA in the nucleus increases by a factor of two, and 3) chromosomes condense as cells enter mitosis.

## 2.3 Sparsity/reconstruction trade-off

Since our model is composed of $\beta$-VAEs, the $\beta$ parameter presents an important trade-off between compact representation and high fidelity image reconstruction. Tuning $\beta$ allows the modeler to preferentially weight the two components of the loss function. A high value of $\beta$ favors a compact representation of the cell in the low dimensional embedding space (i.e. the number of dimensions needed to describe an image $z_r$ and $z_t$) via a higher emphasis on the $KL(q(z|x)|p(z)))$ loss term. A low value of $\beta$ emphasizes accurate reconstruction of the original image by placing more weight on the $\mathbb{E}_{q(z|x)}[\log p(x|z)]$ loss term. The relative emphasis of one term versus the other has consequences for the model and for its applications. For example, one might desire a less-complex data embedding to facilitate the statistical exploration and interpretation of the latent space, while in other circumstances it might be preferable to obtain a more complex embedding that enables the comparative analysis of high-fidelity generated images.

Several methods have been proposed to modulate the trade-off between sparsity and reconstruction in VAEs [24, 25], and other factors such as data normalization, model architecture, hyper-parameters (including training schedules [26]) may also impact this relationship. To demonstrate how our model performs as a function of this relationship, we adopt a reparameterized variational objective that allows us to tune the relative weights of these two terms (see Eq 3).

Because parameter exploration using the full 3D model is prohibitively expensive, we explored the effect of the $\beta$ parameter using a 2D model. This model is the same in all regards to the 3D model, other than that the inputs and convolutions are two dimensional instead of three. Our 2D input data was generated from the 3D data by taking a maximum-intensity projection along the $z$-axis. While this reduction obfuscates some details of the cell's structure and organization, it retains a largely faithful picture of overall cell shape and reconstruction detail, and allows us to explore model and parameter choices approximately an order of magnitude more quickly than using the 3D model.

Using the 2D data, we trained 25 models of cell and nuclear shape ($\mathbf{M_R}$) with $\beta$ values evenly spaced between 0 and 1 using 2D maximum-intensity-projected images of cell and nuclear shape. Using the test data, i.e. data not used in the training phase, we recorded the average of the two terms of ELBO for each of the 25 models and plot the two as a function of $\beta$ in Fig 3A. Sampling from the cell and nuclear shape representation, we show generated images across a range of $\beta$ values in Fig 3C. A few observed cells are visualized in Fig 3B for reference. We note that compared to observed cell and nuclear shapes, generated images close to $\beta \to 0$ retain more detail and perhaps more diversity than images at $\beta \to 1$, although this comes at a trade-off of increased representation dimensionality.

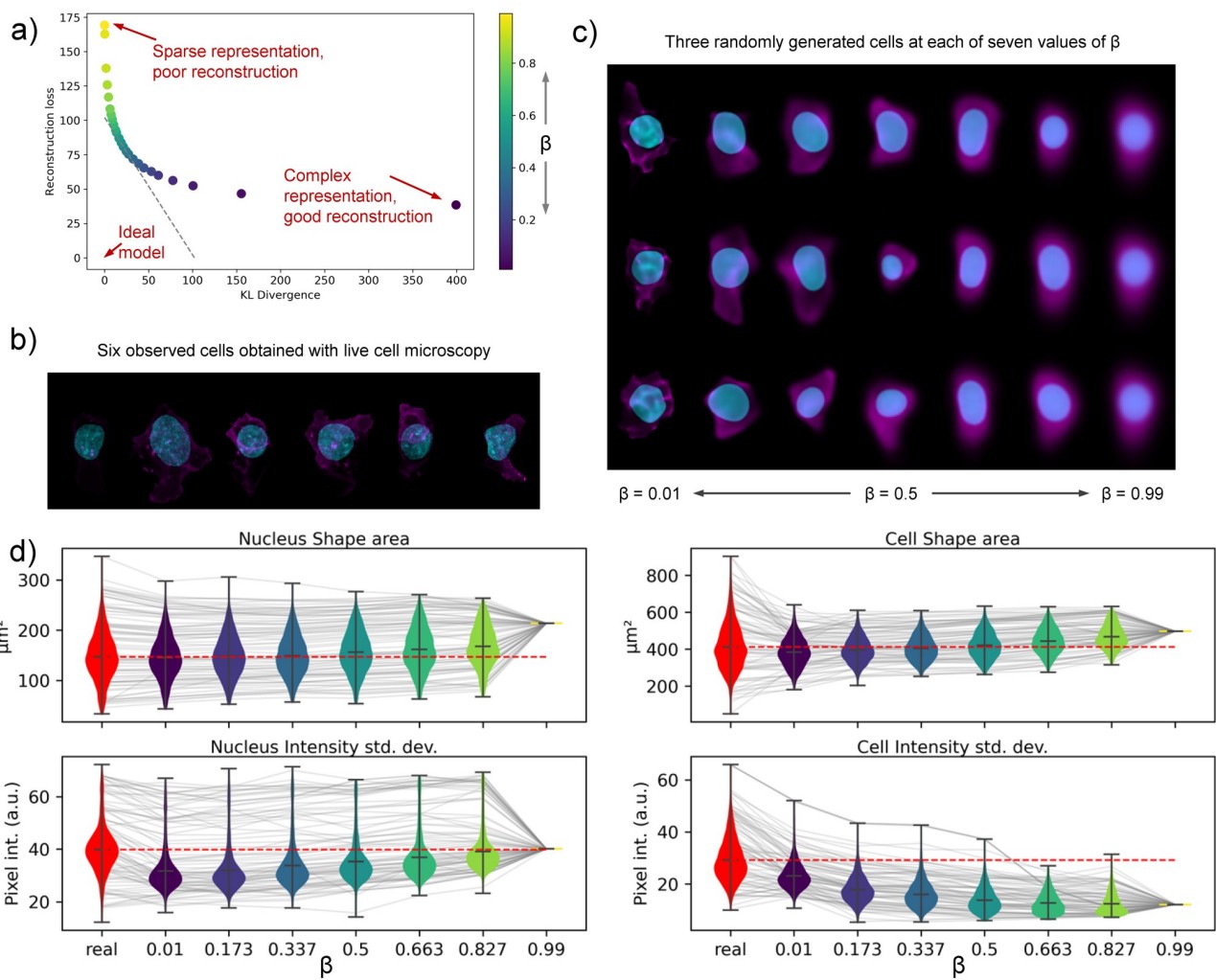

**Fig 3. Evaluation of the sparsity-reconstruction trade-off with 2D Statistical Cell models.** a) shows the average rate and distortion terms of the ELBO for models at a different trade-off $\beta$. Gray dotted line indicates the trade-off achieved by the best performing model. b) Images of six observed cells. Nucleus is in blue. Cell membrane is in magenta. c) Images of cells generated with the Statistical Cell for different values of $\beta$. See S6 Fig for more data on model sparsity. d) Violin plots that show four biologically interpretable features measured on both real (red) and generated cells (blue to yellow, as a function of $\beta$). A grey line shows the feature value for one selected cell; connecting the feature value obtained from the real cell image and the generated cell images. Grey lines are plotted for a subset of all cells used in this analysis.

In order to assess the similarity between real and generated cells beyond visual comparison, we calculated biologically interpretable features for both the real and generated cell images. Specifically, for both the nucleus (channel) and the cell (channel) we obtained three shape features: area, circumference and sphericity, and four intensity-based features: median, mean, standard deviation and entropy. These 14 features were calculated for the approximately 4,000 cells in the test set (See Materials and methods). Results are visualized for four selected features in Fig 3D. Plots for all 14 features are found in S7 Fig. Overall, the shape and intensity features as measured on real cells (red) are highly similar to those obtained on the generated cells images (blue to yellow). The features from the real cells do show more variation than features from generated cells, and surely, models with lower $\beta$ (higher reconstruction) show more similar distributions to those of the real cells than models with higher $\beta$. As $\beta$ approaches 1 (a very

sparse representation with no emphasis on reconstruction), the model generates almost identical images for all the cells. Consequently, the features collapse to a single value.

Additional analyses, detailed in S8 Fig, were performed to see if the information in the latent spaces of the generated models (at different values of $\beta$) were able to recapitulate the 14 features measured from the real cell images. We employed linear regression analyses to statistically explain the first five principal components of these 14 features (amounting to 93% of the total variation among these features) by using the latent space coefficients as input to the regression. These analyses showed that models with a focus on reconstruction (low $\beta$) are able to capture 78% of the statistical variation among the 14 features. However, the drop-off for models with higher $\beta$ is not very steep and even models with very few important latent dimension (#dims $< 10$, $\beta > 0.5$) can still explain more than (60%) variation among these cell features.

## 2.4 Visualization of generated cells and conditionally generated structures

An important application of the trained Statistical Cell model is to visualize cellular structures by generating images sampled from the probabilistic models. That is, by sampling from the latent space that describes the trained probabilistic model of structures dependent on cell and nuclear shape ($\mathbf{M_T}$), we can visualize the location and shape of subcellular structures, and how those properties vary in the data. Moreover, the construction of the model enables us to predict and visualize multiple subcellular structures in the same cell geometry simultaneously, whereas the data the model is trained on only contains one structure tagged per image. Because the model is probabilistic, it approximates the diversity of structure localization specific to each structure type. Fig 4B depicts multiple examples of different subcellular localization patterns given an observed cell and nuclear shape.

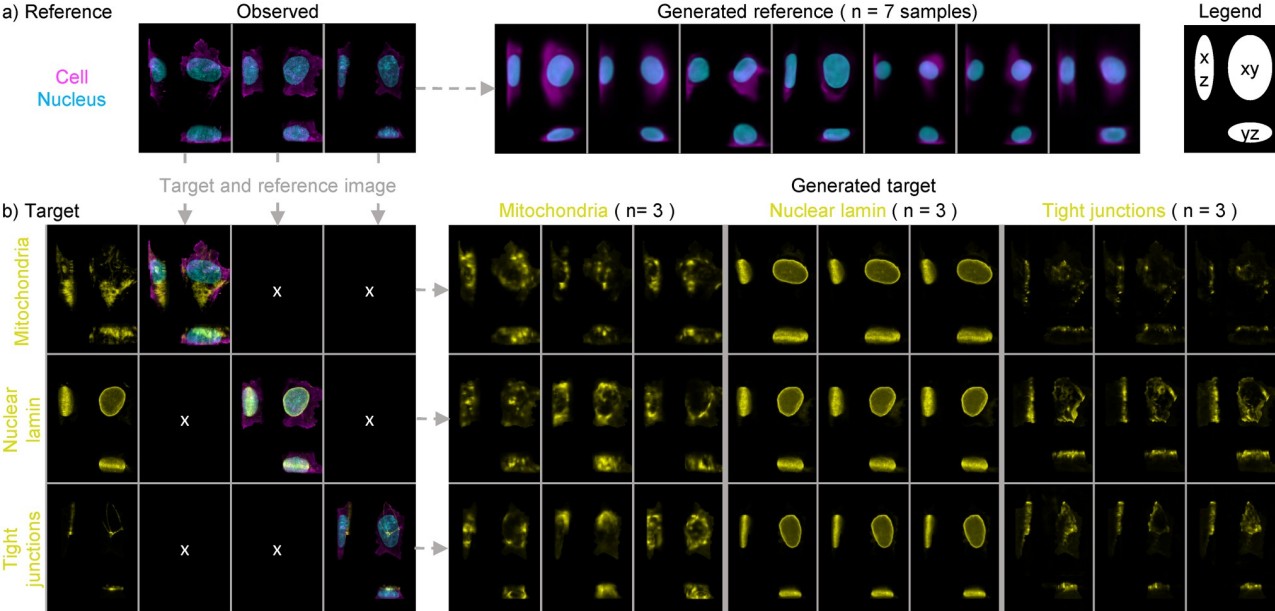

**Fig 4. Generating cell images from the Statistical Cell model.** a) Images can be generated from the probabilistic model of the cell membrane and the nucleus. On the left are three examples of actual cells. On the right are seven generated cells, sampled using independent random draws from the latent space. b) Cell and nuclear image data of actual cells can be used to generate a simulated image of a given structure even if that structure was not measured for that cell. On the left are three actual cells for which three different structures were imaged. From top to bottom: mitochondria, nuclear envelope and tight junctions. On the right are depictions of the generated structure channels for each of these cells and structures. Three examples of each structure in each cell are shown, each generated using independent random draws from the latent space. See S4 and S5 Figs for more examples.

Overall, we find that in distribution the generated structures vary in localization as one would expect, providing strong evidence that the network is successfully learning appropriate rules governing structure-specific localization that are not explicitly encoded in the images themselves or in the "target type" input. For example,

- Mitochondria are distributed throughout the cytoplasm, but are never found inside the nucleus.

- The nuclear envelope forms a closed shell around the DNA.

- The tight junctions are at the apical surface of the cell and around the cell periphery.

- The nucleoli (both the Dense Fibrillar Center and the Granular Component) form blobs that are always inside of the nucleus, never outside.

- Matrix adhesions are always at the basal surface.

The latter two examples along with other examples of typical organelle localization are found in S5 Fig. It is important to note that due to the limitations of the data and the specific model construction, generated subcellular structures are independent of each other (e.g. generated tight junctions may overlap with generated mitochondria). We also observed that across structures there is a great variation in similarity between instantiations generated from the same cell and nuclear geometry. In Fig 4B, we see there is little variation between the three generated instantiations of the nuclear envelope, and these instantiations are very similar to the real cell image. Surely, the nuclear envelope is relatively easy to predict from the nucleus (DNA stain) channel; it's predicting the envelope of a three dimensional (fluorescent) object. But the mitochondria, which are certainly much more uncoupled (biologically and statistically) to the nucleus and cell membrane, and more varying in overall shape and show widely different instantiations.

## 2.5 Quantification of the coupling of subcellular structure localization to gross cellular morphology

In the previous sections we aimed to show, both qualitatively and quantitatively, that the Statistical Cell enables us to model the organization of subcellular structures by leveraging the reference channels, i.e. the cell membrane and the nuclear shape. An important next question is to what extent the reference channels by themselves inform the prediction of subcellular structure organization.

To answer this question, we constructed a measure of coupling sensitivity between a subcellular structure and the morphology of either the cell membrane or the nucleus. Specifically, we quantified the sensitivity of our model to the coupling between each subcellular structure and a reference channel (say, cell membrane) by comparing the ELBO for that unperturbed image with the ELBO of a perturbed version of that image, where the reference channel (e.g. membrane) is replaced by a randomly selected membrane channel from the other cells in the population (see Materials and methods, Eq 4).

As controls for this metric, we created artificial "structure" channels that are duplicates of each of our reference channels (cell membrane and nucleus) to confirm that this measure of coupling makes sense in the limit of structures that are perfectly correlated with one of the reference channels. Fig 5A shows the coupling metrics across three subcellular structures as well the controls. We observed that each control is quantified as being coupled only to itself, and not to the other reference channel; that is, it appears that the cell membrane and the nucleus are not informative of each other under our model. We also used a blank structure as a control,

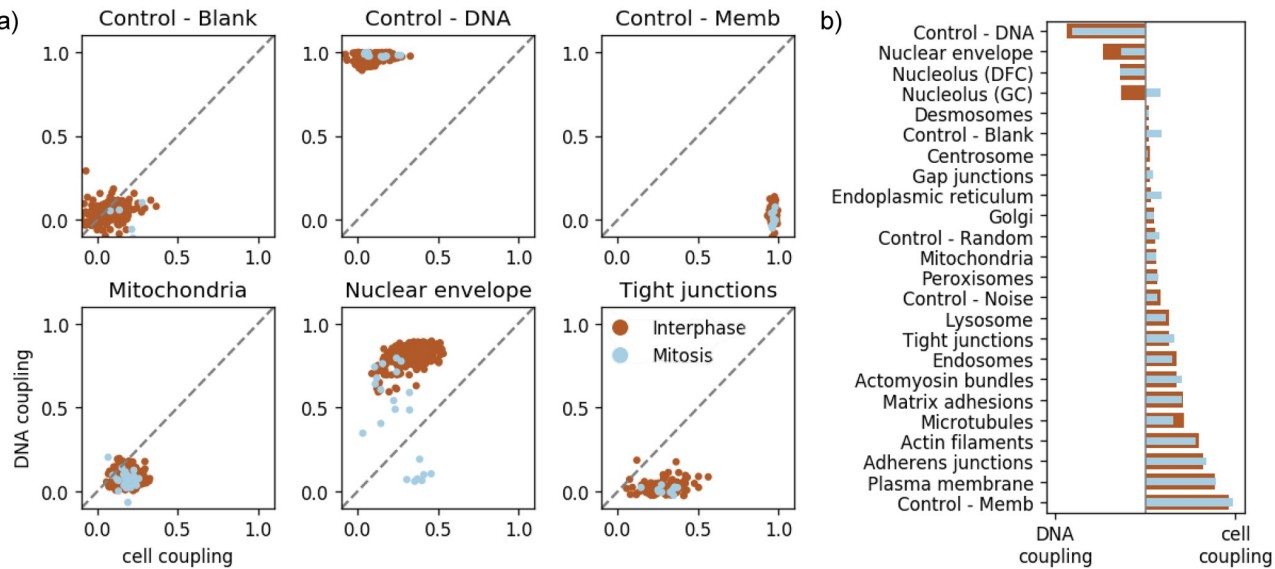

**Fig 5. Quantification of the coupling of cellular morphology and subcellular structure.** a) shows the relative coupling strength of three structures to the nuclear shape (y-axis) and cell membrane (x-axis) of the cells in which they reside, according to Eq 4. Each point represents a cell; brown points are cells in interphase, blue points are cells undergoing mitosis. b) shows the relative degree of coupling of each structure to the cell membrane or nuclear channel, and how this changes during mitosis.

and as expected did not see any coupling between it and either reference channel. Regarding the three subcellular structures: As described above, instantiations of mitochondria from the model are highly variable even within a single given cell and nuclear reference shape, so it is to be expected that they show relatively low coupling to both the nuclear and cell membrane channels. For the nuclear envelope, satisfyingly the cells in interphase demonstrate tight coupling to the nuclear reference channel, however this coupling decreases as cells enter mitosis, when the nuclear envelope is disassembled. Finally, the tight junctions in hiPS cells are confined to a narrow band at the very top of the lateral sides of the cell, and so are expected to show reasonable coupling to the plasma membrane reference channel, but no coupling at all to the nuclear reference channel.

In order to understand how strongly each structure is coupled to the cell or nuclear reference structure overall, in Fig 5B we condensed the coupling metrics across many cells, as displayed in Fig 5A, to a single summary statistic for each of the 24 subcellular structures. The statistic that we employed is the normalized difference of the nuclear coupling to the cell membrane coupling from Eq 4, averaged over all cells in that structure/phase of the cell cycle (see Eq 5 for details).

Fig 5B shows a spectrum of differential couplings under our model, all of which are biologically plausible. As expected, the plasma membrane marker (GFP fused to the CAAX domain of K-Ras) shows the strongest coupling to the reference channel derived from the membrane dye. Both the adherens junctions and the actin filaments in hiPS cells are highly enriched along the lateral sides of the cells, where they make contacts with their neighbors, and so it is expected that they also show strong coupling with the membrane reference channel. At the other end of the spectrum, the nuclear envelope, as described above, is expected to show the best coupling with nuclear shape, followed by the two nucleolar compartments (DFC and GC). The granular compartment of the nucleolus (GC) disassembles during mitosis, and this is readily shown by the loss of coupling to the DNA reference channel. The many structures that

show a range of weak couplings to the reference channels include punctate structures that are present in only a few copies per cell including centrioles, desmosomes and gap junctions. Some structures that effectively fill up most of the cytoplasm (including microtubules and endoscopes) typically show stronger coupling to the membrane reference channel than to the nuclear reference channel, though these couplings are never as strong as for the actual membrane-associated structures.

## 2.6 Evaluation of drug perturbation effects on subcellular structures

As a small pilot experiment, we evaluated the model's ability to detect morphological perturbations using two well-characterized drugs with known structural targets: Brefeldin A which causes disassembly of the Golgi apparatus, and paclitaxel (Taxol) a chemotherapy medication used for a variety of cancers that stabilizes microtubules and leads to microtubule bundle formation. For cells treated with each drug, we imaged both the target structure (Golgi for brefeldin and microtubules for paclitaxel, respectively) as well as a distinct structure that is not expected to be perturbed by either drug (tight junctions). The drug dataset was collected under the same experimental conditions, and using the same microscope and same cell lines as the for the main dataset that the Statistical Cell model was trained on. Low drug concentrations were used (5.0 $\mu$M in both cases) to fall in a range where the target structures were clearly perturbed without any overall drastic change in cell shape. See Fig 6A and 6B and Section 4.1.4.

Latent embeddings (reference $z_r$ and target $z_t$) were computed for each image of drug-treated cells and vehicle-treated controls by running them through the trained Statistical Cell model. Thus, no new training was performed on the drug perturbation data. Reference latent embeddings $z_r$ were calculated by running the image data of the cell and nuclear channels through the trained model of cell and nuclear shape, $\mathbf{M_R}$. Calculating the conditional latent embeddings $z_t$ using the trained target model $\mathbf{M_T}$ requires three data elements: 1) the reference channels (cell and nucleus), 2) the target channel (GFP-tagged structure), and 3) the selector variable $t$ indicating one of the 24 organelles which in case of the drug perturbation dataset is either Golgi, microtubules or tight junctions. See the three arrows in Fig 1C moving into the encoder block and from there into the conditional latent space $z_t$.

Fig 6C shows the first two dimensions (as ranked by mean absolute deviation) of this embedding for the reference channels. In this original reference latent space (defined by training the Statistical Cell model on the large collection of more than 40,000 cells), both drug-treated cell populations (blue and pink dots) as well as the vehicle-treated control cells (black dots) from this pilot experiment showed overlapping distributions in the top two latent dimensions that are centered around the origin. This observation confirms that the chosen drug concentrations have little or no effect on overall cell shape.

However, strong effects are clearly observable in the latent spaces corresponding to the targets of the two drugs. Specifically, the microtubule latent space embeddings for the paclitaxel-treated cells show a significant shift in the latent space positions of the overall population, such that the centroid of the population of drug-treated cells is far removed from the latent space origin (Fig 6D). Indeed, this population shift corresponds to a clear, systematic shift in microtubule distribution from the dispersed, near-uniform distribution characteristic of untreated cells to a distinct bundled phenotype (Fig 6G). Similarly, for the brefeldin treatment, the distribution of treated cells in the Golgi latent space is significantly altered, with some cells remaining close to the origin but a new subset of cells appearing to be strongly shifted in the most significant dimension (blue dots in Fig 6E). We also imaged the distribution of the Golgi apparatus for a handful of palictaxel-treated cells (pink dots in Fig 6E); as expected, these embeddings fully overlap with the vehicle-treated control cells. Finally, in the latent space of the tight

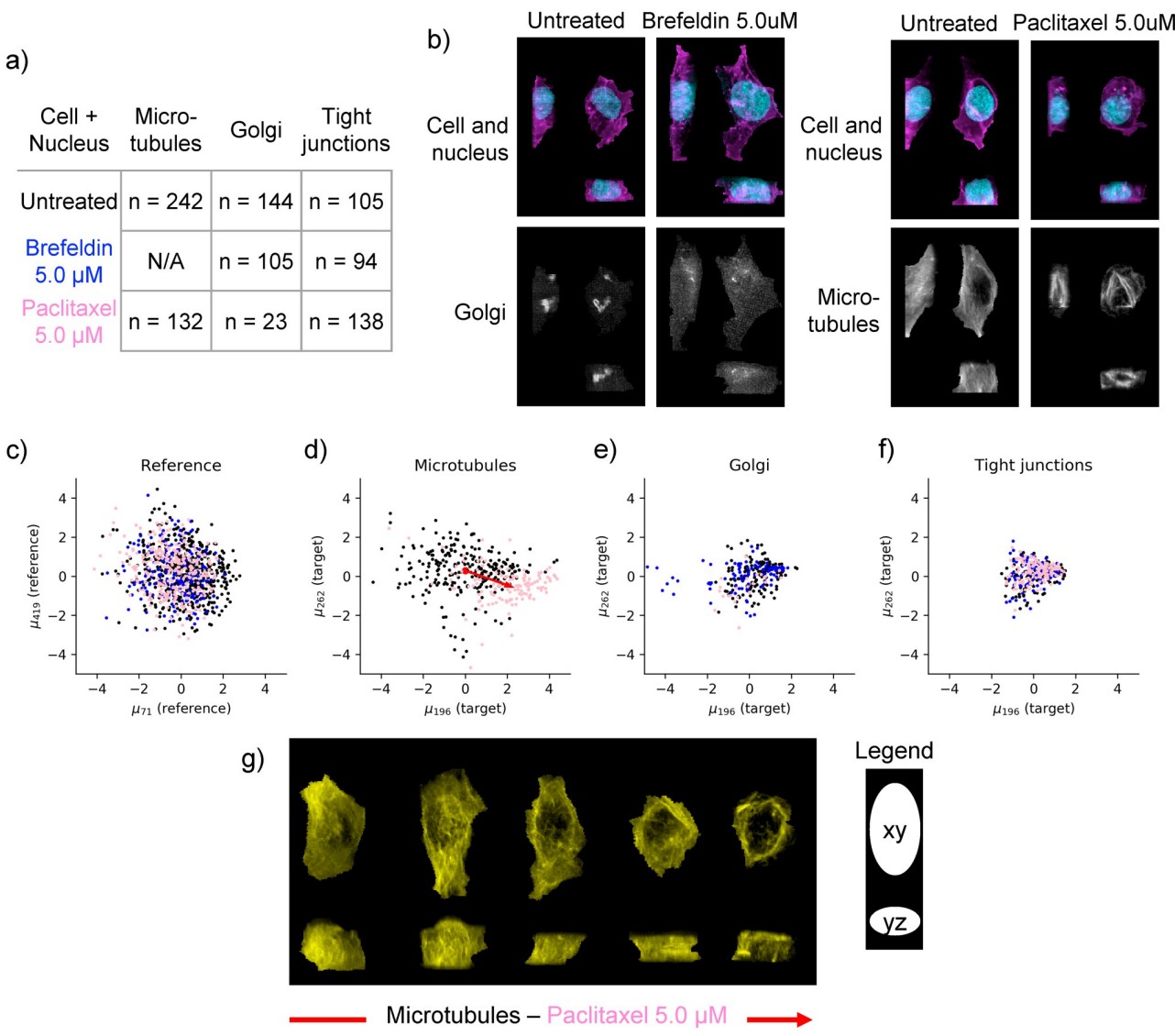

**Fig 6. Analysis of drug perturbation data with the Statistical Cell.** a) Experimental overview of drug-treated cells (rows) and measured subcellular structures (columns) where n indicates the number of obtained cells. Drugs are color-coded (Brefeldin is blue, Paclitaxel is pink). b) Maximum intensity projections of four real cells that are part of the drug perturbation study. For each cell the reference channels (nucleus in blue, cell membrane in magenta.) are visualized in the top, and the structure channel in the bottom. On the left: An untreated and Brefeldin-treated Golgi-tagged cell is shown. On the right: An untreated and Paclitaxel-treated Microtubules-tagged cell is shown. c) Scatter plot of cells embedded in first two dimensions of the reference latent space (cell + nucleus) of the Statistical Cell model. d) Microtubules-tagged cells embedded in the first two dimensions of the conditional latent space. The red arrow indicates the direction of the change of the centroid of the untreated (black) and treated (pink) populations. e) and f) Similar to d), but for Golgi and tight junctions, respectively. g) Visualization of microtubules for five real cells. For each cell we have visualized the maximum intensity projection along z, i.e. looking at the xy plane (top), and the maximum intensity projection along x, i.e. looking at the yz plane (bottom). These five cells are sampled at fixed intervals from the untreated(left)-to-treated(right) direction after projecting onto the red line in d).

junctions all three populations (both drug treatments and the vehicle control) show complete overlap in a symmetric distribution near the origin, consistent with the expectation that these drug treatments should not significantly alter the organization of tight junctions in these cells (Fig 6F). Importantly, the results of this pilot experiment suggest that the model is capable of producing reasonable latent space embeddings for structures that are outside of the range of

the original training set (specifically the microtubules for paclitaxel-treated cells and the Golgi for brefeldin-treated cells).

## 3 Discussion

The Statistical Cell is a model of the three-dimensional organization of subcellular structures in human induced pluripotent stem cells. We have described the model and its capabilities, and explored how it may be a useful tool for studying the organization of cells from fluorescent 3D spinning disk confocal images. In this section, we discuss important considerations and future work of the Statistical Cell.

Previous incarnations of our Statistical Cell model [27, 28] featured an adversarial loss function. Along with non-generative direct image transformation methods, e.g. [8, 10], generative adversarial networks (GANs) have been proposed to produce high-fidelity generated cell images [19, 29]. Although these adversarial-loss based methods may produce crisp images when appropriately tuned, they remain notorious in their difficulty to efficiently optimize [30, 31], suffer from several poorly understood and difficult to diagnose pathologies such as "mode collapse" [32] and difficulty to initialize models that produce large images [30, 31].

It should be noted that both the VAE loss function used here and adversarial autoencoders [33] allow for the construction of a low dimensional representations of specified distribution (e.g. Gaussian-distributed latent spaces). We ultimately chose a VAE-based model version for a number of reasons, both practical and theoretical. Compared to GANs, $\beta$-VAEs are more stable and easier to train, possess stronger theoretical bounds [23], and are able to easily trade off latent-space dimensionality for reconstruction fidelity [24] while also producing information-independent latent representations. In this study we observed that the marginal distributions of individual latent space dimensions were more irregular for GANs, and more normal for $\beta$-VAEs (see S1 Fig) enabling a smooth "latent space walk" (see S2 Fig), and overall, a more straightforward way to interpret the latent dimensions.

In this work we employ a standard conditional $\beta$-VAE architecture, composed of convolutional/residual blocks. We present no drastic architectural innovations, but rather leverage the proven reliability of $\beta$-VAEs in other domains to build a solid framework for integrating and modeling 3D spinning disk confocal fluorescent microscopy data. Our focus is the cell biological application, and the way that the $\beta$-VAE enables hands-off quantification of image data that are difficult to describe in anything other than qualitative terms.

There are a number of ways in which our model could be extended in the future. In this work, we showed that accurate shape features of cells can be derived from Statistic Cell model. A next step would be to incorporate such features in the model formulation. Recent work in non-biological domains has suggested that using image features and auxiliary loss functions for VAEs can improve model performance and generalizability [34]. Specifically, directly optimizing our model to retain specific morphological properties, e.g. using feature-based losses such as cell volume, cell height, etc., may improve image generation while retaining specific statistical properties relevant to the application of these models.

As in many bioimage informatics applications, image pre-processing and normalization are important issues and can have important consequences to downstream workflows. In the Statistical Cell model, image normalization and the loss function (ELBO) are coupled to each other. Changing the way in which the cell images are pre-processed directly impacts how the loss function affects the training of the model. For instance, noise and bright spots in the cell and nuclear dyes are penalized disproportionately to their biological significance. We have not thoroughly studied different normalization schemes and their effects on the inferred model,

and as with many image analysis methods, it is likely that results may vary as a function of image pre-processing.

One avenue of approach to normalization could be to use image segmentations as inputs to the model, rather than the fluorescent dye intensities. Segmentation itself presents its own problems, but as 3D segmentation techniques for densely packed spinning disk confocal fluorescent microscopy images improve [6], this approach should become more feasible and deserving of serious consideration. Our framework is amenable to incorporating both segmentation-based inputs and hand-crafted features, a future direction that holds promise for incorporating the most desirable facets of each of these approaches.

An important aspect of the Statistical Cell model is the application of the trained model to external data. Although our model may be useful for determining the effects of drug-perturbations, as with any model it is important to understand to what extent it may be applied in new contexts. The results presented here were trained on a data set of relatively few conditions, and we do not expect our model to generate realistic images or produce accurate representations of cellular structures on images that were collected in a substantially different way, e.g. different cell types, different microscopes, etc., although future models may be able to account for these types of biological and technical variation [35].

It will be important to understand whether the Statistical Cell can, at least, be used as a pretrained model to enable faster training on smaller data sets of cell images that were obtained under different conditions. The computational complexity of training (from scratch) a 3D Statistical Cell model requires sizable GPU resources. Specifically, training the 3D Statistical Cell model on a corpus of 40,000 relatively large 3D images as explained in this work required 2 weeks of training time using two 32GB GPUs. Noteworthy, we allowed the model to train until convergence based on visual inspection of ELBO metrics and generated cell images. The use of an automated stopping criterion may reduce compute time.

The Statistical Cell can visualize multiple subcellular structures simultaneously at a single-cell level. Yet, unobserved complex interrelationships between structures may not be accurately represented by the model. This is because the Statistical Cell model captures the partial correlation between a target subcellular structure and the cell and nucleus, but not directly between different subcellular structures. By construction, these components of our model (the image data of 19 subcellular structures) are modeled independently of one another. Through the microscope these 19 subcellular structures come in different shapes, sizes, locations and number of copies. Also, imaging aspects, such as intensity distributions, vary substantially across the 19 subcellular structures. This complicates the interpretation of the latent spaces in a shared manner across the subcellular structures. Cell lines that have been gene-edited to express multiple tagged subcellular structures could be leveraged to model a richer covariation structure. The Allen Cell Collection contains various such cell lines. Beyond jointly modeling more simultaneously acquired image channels, additional data types (e.g. RNA FISH) should be incorporated into a generative single-cell model. Conditional modeling and visualization of multiple jointly acquired data modalities is necessary to move towards a truly integrated picture of cell state.

## 4 Materials and methods

### 4.1 Data

Our model was trained on publicly available cell image data generated at the Allen Institute for Cell Science.

**4.1.1 Allen cell collection.** The input data for training can be obtained at allencell.org or directly at https://open.quiltdata.com/b/allencell/packages/aics/pipeline_integrated_single_

cell. This image resource is part of our ongoing efforts at the Allen Institute for Cell Science to image and analyze subcellular structures in human induced pluripotent stem cells. Imaging and culture conditions are described in [8]. Each source image consists of channels corresponding to the reference nuclear signal and cell membrane signal, and a fluorescently labeled target sub-cellular structure. Extensive information can be found on allencell.org.

**4.1.2 Preprocessing of images.** All cell regions were segmented from the field of view using a segmentation workflow. See https://www.allencell.org/extracting-information.html for more details. Each channel was processed by subtracting the most populous pixel intensity, zeroing-out negative-valued pixels, and re-scaling image intensity to a value between 0 and 1. The cells were aligned by the major axis of the cell shape, centered according to the center of mass of the segmented nuclear region, and flipped according to image skew. Each of the 49,340 cell images were linearly rescaled to cubic voxels of 0.317 μm/px, and padded to $128 \times 96 \times 64$ cubic voxels. 2D images for Section 2.3 were created by maximum-intensity projection along the $z$-axis, independently for all three image channels, and are available in the data package.

**4.1.3 Mitotic annotations.** Cells were annotated as being in interphase or mitosis via manual inspection of images by a resident expert cell biologist and released as part of the Allen Institute—Integrated Mitotic Stem Cell. Mitotic cells were further annotated into four classes based on the phase of mitosis they were in: 1) prophase, 2) early prometaphase, 3) prometaphase / metaphase, and 4) anaphase / telophase / cytokinesis. Further details are available at https://www.allencell.org/hips-cells-during-mitosis.html#sectionMethods-for-mitotic-cells.

**4.1.4 Drug perturbations.** The data for the drug perturbation experiments were acquired independently of the main training and testing data for the model, and here we describe its acquisition and processing.

A subset of mEGFP-tagged human induced pluripotent stem cells (hiPSCs) from the Allen Cell Collection were selected in this study, including TUBA1B line:AICS-12, ST6GAL1 line: AICS-25, TJP1 line:AICS-23, to show the location of a particular cell organelle or structure and represent cellular organization. mEGFP-tagged hiPSCs were seeded onto Matrigel-coated 96-well plates at a density of 2,500 to 3,500 cells per well and maintained in culture in phenol-red free mTeSR1 media with 1% penicillin streptomycin for 4 days before imaging (media changed every day). On day 4, cells on the 96-well plate were treated with one of the pre-selected, well-characterized drugs with concentration and incubation time described in Table 2.

At the end of the incubation time, hiPSCs were then incubated in imaging media of phenol red-free mTeSR1 media (Stem Cell Technologies) with 1% penicillin streptomycin (Thermo Fisher) with X1 Nuc Blue Live (Hoechst 33342, Thermo Fisher) for 20 min and 3X CellMask (Thermo Fisher) for 10 min. The cells were washed with fresh imaging media prior to being imaged live at high magnification in 3D.

3D Live-cell imaging of mEGFP-tagged hiPSCs was performed on a Zeiss spinning-disk microscope with a 100x/1.2 NA W C-Apochromat Korr UV-vis IR objective (Zeiss) and a 1.2x tube lens adapter for a final magnification of 120x, a CSU-x1 spinning-disk head (Yokogawa) and Orca Flash 4.0 camera (Hamamatsu) (pixel size 0.271 $\mu$m in x-y after 2x2 binning and

**Table 2. Drug treatments.**

| Perturbation agent | Vendor | Catalog | Well-known action | Concentration | Incubation time |
|---|---|---|---|---|---|
| Brefeldin A | Selleckchem.com | No.S7046 | Vesicle trafficking inhibitor | 5 μM | 0.5 hr |
| Paclitaxel | Selleckchem.com | No.S1150 | Microtubule polymer stabilizer | 5 μM | 2 hr |
| DMSO | Sigma Aldrich | N/A | Vehicle control | 0.01% | 0.5 to 2 hr |

0.29 $\mu$m in Z). Standard laser lines (405, 488, 561, 640 nm), primary dichroic (RQFT 405, 488, 568, 647 nm) and the following Band Pass (BP) filter sets (Chroma) were used for fluorescent imaging: 450/50 nm for detection of Nuc Blue Live, 525/50 nm for detection of mEGFP, and 690/50 nm for detection of CMDR dye (Thermo # C10046). Cells were imaged in phenol red-free mTeSR1 media, within an incubation chamber maintaining 37˚C and 5% $CO_2$. Bright field images were acquired using a white light LED with broadband peak emission using the mEGFP BP filter 525/50 nm for bright field light collection.

To obtain single-cell measurements, fluorescent images of cells treated with CellMask Deep Red dye and NucBlue Live dye were segmented with the aforementioned segmentation workflow. The fluorescent images are normalized with a median filter and adaptive local normalization. Nuclei are segmented with active contouring. Segmented nuclei are used to create seeds for segmentation of individual cells based on the signal from the plasma membrane. The plasma membrane signal is boosted at the top of the cell and fluorescent endocytic vesicles are removed from normalized cell membrane image and are then segmented with 3D watershed with seeds from nucleus segmentation.

Further details are available at https://www.allencell.org/drug-perturbation-pilot.html. The data are available at https://www.allencell.org/data-downloading.html#sectionDrug SignatureData.

## 4.2 Model architecture

At its core, the Statistical Cell is a probabilistic model of cell and nuclear shape conjoined to a probability distribution for the localization of a given subcellular structure conditional on cell and nuclear shape. An observed image, $x_{r,t}$, is modeled as $p(x_{r,t}|t) = p(x_r)p(x_t|x_r, t)$, where $r$ indicates *reference* image channels that contain the same cellular structure across all images (in this case the plasma membrane using CellMask Deep Red dye, and the nucleus via labeling DNA with NucBlue Live dye), and $t$ indicates *structure* channels. The model maps $x_r$ and $x_t$ to learn low dimensional variables (or "embeddings"), $z_r$ and $z_t$ that capture morphological variation and relationships between the reference and target structures, allows for sampling of missing data and image exemplars, and characterizes statistical relationships between the reference and target structures.

The Statistical Cell model consists of two sub-models that are trained (conditionally) independently, the model of cell and nuclear shape, $\mathbf{M_R}$, and the conditional model of structure localization, $\mathbf{M_T}$.

A diagram of the model is shown in Fig 1B and 1C. The reference component $\mathbf{M_R}$ consists of an encoder that computes the variational posterior, $q(z_r|x_r)$ constructed by serial residual blocks (see Fig 7) that perform convolutional operations, spatially downsampling the image by half and increasing channel dimension at each layer. The output is then reshaped to a vector and passed independently through two fully connected layers to result in $z_r = N(\mu_{z_r}, \sigma_{z_r})$. $z_r$ is sampled from that normal distribution and passed through a fully connected layer, and passed through residual blocks that spatially upsample, and decrease channel dimension, progressively decoding the latent representation. The same architecture is used for the target model, $\mathbf{M_T}$, but instead the target label and a linearly downsampled copy of $x_r$ is passed in as well.

The primary layer component of this model is a modified residual layer [37], and a detailed description can be found in Fig 7. Table 3 shows the high-level model architecture.

The 2D model was implemented the same as above but with 2D convolution operations. The number of parameters for the 3D and 2D models are 122,627,829 and 22,279,054, respectively.

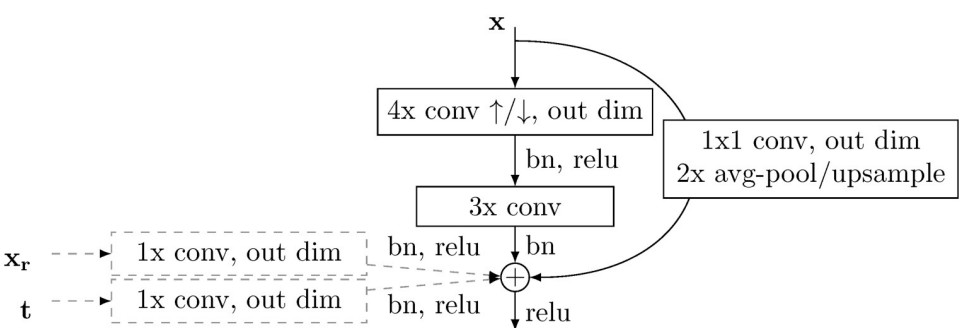

**Fig 7. Residual block used in this model.** Each layer of our model is a modified residual layer. In the encoder, the layer input, **x**, is passed through a 4x convolution kernel with a stride of 2, then a 3x convolution kernel with a stride of 1 or a 1x convolution kernel with a subsequent avg-pooling step. These results are summed along the channel dimensions, added, and passed to the next layer. With the decoder, 4x convolution is replaced with transposed convolution, and pooling replaced with linear upsampling. In the case of the conditional model (components with dotted lines) $\mathbf{M_T}$, the reference input $x_r$ is linearly interpolated to be the same size as the output, and passed through a 1x kernel. The target label is passed through a 1x kernel, and added to each pixel of the output. Spectral weight normalization [36] is utilized at every convolutional or fully-connected operation. In the case of the 3D model the convolutions are three-dimensional, and the 2D model uses two-dimensional convolutions.

The model is trained to maximize the Evidence Lower Bound (ELBO) given an input image $x_{r,t}$:

$$\log p(\boldsymbol{x}_{r,t}|t) \geq \text{ELBO}(\boldsymbol{x}_{r,t}|t) = \mathbb{E}_{q(z_{r,t}|x_{r,t})}[\log p(\boldsymbol{x}_{r,t}|\boldsymbol{z}_{r,t},t)] - \text{KL}(q(\boldsymbol{z}_{r,t}|\boldsymbol{x}_{r,t},t)|p(z)) \quad (1)$$

An interpretation of this procedure is that we seek to find a model such that the observed data is the most probable under the model distribution, with the ELBO is as (tractable) approximation of this quantity. Under the generative model described in Fig 1B, we factor out

**Table 3. Architecture of model used in this study.** Arrows indicate spatial downsampling or upsampling.

| Reference Model, $\mathbf{M_R}$ | Target Model, $\mathbf{M_T}$ |
|---|---|
| $x_r$ | multicolumn1c $x_t$ |
| Residual layer, ↓, 32 | Residual layer + $(x_r, t)$, ↓, 32 |
| Residual layer, ↓, 64 | Residual layer + $(x_r, t)$, ↓, 64 |
| Residual layer, ↓, 128 | Residual layer + $(x_r, t)$, ↓, 128 |
| Residual layer, ↓, 256 | Residual layer + $(x_r, t)$, ↓, 256 |
| Residual layer, ↓, 512 | Residual layer + $(x_r, t)$, ↓, 512 |
| [FC 512], [FC 512] | [FC 512], [FC 512] |
| $z_r = N(\mu_{z_r}, \sigma_{z_r})$ | $z_t = N(\mu_{z_t}, \sigma_{z_t})$ |
| FC 512 | FC 512 |
| Residual layer, ↑, 512 | Residual layer + $(x_r, t)$, ↑, 512 |
| Residual layer, ↑, 256 | Residual layer + $(x_r, t)$, ↑, 256 |
| Residual layer, ↑, 128 | Residual layer + $(x_r, t)$, ↑, 128 |
| Residual layer, ↑, 64 | Residual layer + $(x_r, t)$, ↑, 64 |
| Residual layer, ↑, 32 | Residual layer + $(x_r, t)$, ↑, 32 |
| Residual layer, ↑, 2 | Residual layer + $(x_r, t)$, ↑, 1 |
| $\hat{x}_r$ | $\hat{x}_t$ |

structure and reference components and train two separate components:

$$\log p(\boldsymbol{x}_{r,t}|t) = \log p(\boldsymbol{x}_t|\boldsymbol{x}_r, t) + \log p(\boldsymbol{x}_r) \geq \mathrm{ELBO}(\boldsymbol{x}_t|\boldsymbol{x}_r, t) + \mathrm{ELBO}(\boldsymbol{x}_r) \qquad (2)$$

The embeddings produced by the encoder $q(z|x)$ are encouraged to be compact in the sense that they are penalized for differing in distribution from a standard normal distribution (as measured by $\mathrm{KL}(q(z|x)|q(z))$). An embedding, however compact, is only useful insofar as it is able to faithfully recapitulate the data. The decoder $p(x|z)$ is optimized to produce faithful reconstructions via the reconstruction error term $\mathbb{E}_{q(z_{r,t}|x_{r,t})}[\log p(\boldsymbol{x}_{r,t}|\boldsymbol{z}_{r,t})]$, which encourages the model to balance compactness against transmitting enough information to accurately reconstruct the data.

For section 2.3 we adopt the following reparameterized ELBO definition:

$$\mathrm{ELBO}(x) = (1 - \beta)\mathbb{E}_{q(z|x)}[\log p(x|z)] - \beta KL(q(z|x)|p(z)). \qquad (3)$$

where $\beta$ is between 0 and 1.

This slight modification to the objective function allowed us to trade-off the relative importance between the reconstruction and sparsity terms of our loss function while keeping the order of magnitude of the total objective function constant. This is greatly beneficial in training multiple models at different values of $\beta$, without needing to modify any other hyper parameters to compensate for an objective function that grows with $\beta$, as in the traditional parameterization of the $\beta VAE$ objective function.

## 4.3 Calculation of Evidence Lower Bound

To calculate the ELBO we use the low-variance estimator in [23]. We use pixel-wise mean squared error to approximate the reconstruction likelihood and average over ten samples from $\boldsymbol{z}_r$ or $\boldsymbol{z}_t$.

## 4.4 Training procedure

Each model is trained with a batch size of 32 at a learning rate of 0.0002 for 300 epochs via gradient-descent with the Adam optimizer [38]. The optimizer $\beta$ hyper-parameter values are set to (0.9, 0.999) (not to be confused with $\beta$ in the model's objective function). The maximum latent space dimensionality for the reference structures, $\boldsymbol{Z}^r$, and target structures, $\boldsymbol{Z}^s$, were each set to 512 dimensions. We adopt the stochastic training procedure outlined in [23].

We split the data set into 80% training, 10% validation and 10% test, and trained both the reference and conditional model for 300 epochs, and for each training session use the model with the highest ELBO on the validation set.

The model was implemented in PyTorch version 1.2.0, and each component ($P(x_r)$ and $P(x_t|x_r, t)$) was trained on an NVIDIA V100 graphics card. 3D models took approximately two weeks to train while 2D models took approximately 1.5 days to train.

To address overfitting, we evaluate the ELBO on images assigned to the validation set at every epoch. For all results in this manuscript, the model with the highest validation-set ELBO is used. For sparsity/reconstruction models in Fig 3, we use the unweighted ELBO.

## 4.5 Cell feature extraction and feature space analysis

Real cell and nucleus images from the test set and their generated counterparts are segmented prior to shape- and intensity-based feature extraction (Fig 3D and S7 and S8 Figs). Generated images from each 2D model (with different values of $\beta$) are segmented using adaptive thresholding with an arithmetic mean filter and a block size of 199. All pixel values are

scaled to fit within an 8-bit intensity range. Then, 3 shape features and 4 intensity features are calculated from the cell and nucleus channel (separately) using https://github.com/AllenCell/aicsfeature.

To evaluate how well the feature space of the real cells maps to the latent spaces of the 2D models with different values of $\beta$, latent space dimensions with the highest mean KLDs are first identified by applying a KLD threshold of 0.6 across the sorted mean KLD (ranked per dimension) across all 2D models (S8(A) Fig). Principal component analysis (PCA) is then applied to the 14 extracted features from the test set to arrive at a feature space, and the top 5 principal components (PCs), with a total explained variance ratio of $> 0.9$, are selected. The explained variance ratio and the loading of each feature for the top 5 PCs are shown in S8(B) and S8(C) Fig. Finally, the "important" latent dimensions, i.e. those above the KLD threshold of 0.6, of the test set embeddings in each 2D model are used to fit a linear model for each of the 5 top PCs independently using linear regression analysis. About 90% of the test set embeddings are used to fit the model, and the remaining 10% is used for prediction and $R^2$ score evaluation. The $R^2$ scores for each linear regression model (each top PC) are plotted as a function of $\beta$ in S8(D) Fig.

### 4.6 Statistic of subcellular structure coupling

The per-channel statistic we display in Fig 5B is computed for each cell $x_i$ by considering the likelihood of that cell under the model, compared to the likelihood of that cell with one of its channels swapped out for that same channel from a different cell:

$$c_i^{rs} = \frac{\text{ELBO}(x_i = [x_i^r, x_i^{r'}, x_i^s])}{\frac{1}{N_s}\sum_{j=1}^{N_s} \text{ELBO}(x_{ij} = [x_j^r, x_i^{r'}, x_i^s])} \tag{4}$$

Here $r$ is the reference channel (either the membrane or the nucleus) that we are evaluating, $s$ is the structure type, denoting which set of cells we aggregate over. $x_i = [x_i^r, x_i^{r'}, x_i^s]$ is the three channel image decomposed into the reference channel of interest $r$, the other reference channel $r'$, and the structure channel $s$. The numerator is the likelihood of the original data, and the denominator is the average permuted likelihood of the cell with the reference channel of interest $r$ permuted across all other cells with the same structure tagged (i.e. $N_s$ cells).

To aggregate this per-channel coupling strength into a *relative* coupling value, we take the ratio of the difference over the sum of the membrane-structure coupling and the nucleus-structure coupling. That is, the differential coupling of a structure to the membrane vs the nucleus $d_s$ is computed as

$$d_s^p = \frac{1}{N_s^p}\sum_{i=1}^{N_s^p}\frac{c_i^{ms} - c_i^{ns}}{c_i^{ms} + c_i^{ns}} \tag{5}$$

where $N_s^p$ is the number of cells where structure $s$ is tagged and are also in cell cycle phase $p$ (interphase or mitosis), $c_i^{ms}$ is the coupling of structure $s$ in cell $i$ to the membrane, and similarly $c_i^{ns}$ is the coupling of the structure in that cell to the nucleus.

### Supporting information

**S1 Fig. Pairwise plots of the top four latent space dimensions, as ranked by mean absolute deviation from 0 on the test set.** The marginal distribution of each latent dimension is plotted on the diagonal. a) Here we color by the cell volume, and see a visually apparent pattern in the

data—most notably a strong correlation with $\mu_{188}$. b) Here we color by the cell height, and again observe structure in the scatter plots—most notably a strong correlation with $\mu_{419}$.
(TIF)

**S2 Fig. Latent space walks along the 3rd through 6th highest ranked dimensions, as ranked by mean absolute deviation from 0 on the test set.** Walks are performed along the specified dimension in nine steps, starting at negative two standard deviations and ending at two standard deviations. All other latent dimensions are set to 0. We include the name of the most highly correlated cell feature, but the cell features are highly correlated (see S3 Fig) and a single latent space dimension may correlate with many cell features. a) Latent dimension $\mu_{465}$, which is most strongly correlated with nuclear surface area. b) Latent dimension $\mu_{188}$, which is most strongly correlated with cell volume. c) Latent dimension $\mu_{238}$, which is most strongly correlated with tilt/shear along the *x-z*-axes. d) Latent dimension $\mu_{107}$, which is most strongly correlated with the total integrated intensity in the plasma membrane dye channel.
(TIF)

**S3 Fig.** a) Heatmap of Spearman correlations of reference latent space dimensions with single-cell features on all cells in the test set. Cell features are hierarchically clustered. Latent space dimensions are sorted in descending rank by mean absolute deviation from 0, and for clarity only the top 32 dimensions are shown. Dimensions below 32 displayed significantly more noise and less correlation with cell features. b) Mean absolute deviation from 0 of all reference latent space dimensions, sorted by value. Values are computed by averaging over all cells in the test set. c) Explained variance of principal components of the z-scored cell features on all cells in the test set. d) Pearson correlation of the top 32 dimensions of the latent space, computed on all cells in the test set as ranked by mean absolute deviation from 0. We note that these dimensions display little to no correlation structure, empirically verifying the ability of the $\beta$-VAE to produce a disentangled latent space.
(TIF)

**S4 Fig. Three examples of each mEGFP-tagged structure are shown, sampled randomly from our test data set.** Each cell only has one mEGFP-tagged structure, so examples are all from different cells.
(TIF)

**S5 Fig. Structures generated by our model.** Three examples of each mEGFP-tagged structure are shown. Structures are generated using random draws from the conditional latent space, while keeping the reference geometry fixed to a single (randomly chosen) cell geometry from the test set. The same cell geometry is used across all structures shown here.
(TIF)

**S6 Fig.** a) Mean KLD per dimension for the reference latent space of the test set in the 2D model, as a function of $\beta$, averaged over all dimensions in the latent space. b) Mean KLD per dimension for the reference latent space of the test set in the 2D model, as a function of dimension rank, for each model fit using a different $\beta$. c) Left: Mean KLD per dimension for the reference latent space of the test set in the 3D model, as a function of dimension rank. Right: Mean KLD per dimension for the conditional latent space of the test set in the 3D model, as a function of dimension rank and structure type.
(TIF)

**S7 Fig. Violin plots that show fourteen biologically interpretable features measured on both real (red) and generated cells (blue to yellow, as a function of $\beta$).** Plots on the left show seven features based on the nucleus (channel); plots on the right show the same seven features

based on the cell (channel). These seven features include three shape features: area, circumference and sphericity, and four intensity-based features: median, mean, standard deviation and entropy. A grey line shows the feature value for one selected cell; connecting the feature value obtained from the real cell image with the values obtained from the generated cell images. Grey lines are plotted for a subset of all cells used in this analysis.
(TIF)

**S8 Fig.** a) Ranked mean KLD per dimension for the reference latent space of the test set in selected 2D models trained with different values of $\beta$. Dimensions with a KLD larger than 0.6 are considered 'important' latent space dimensions. Models with low $\beta$ (focus on reconstruction) have more important latent space dimensions than models with high $\beta$ (focus on sparsity). b) Explained variance ratios for the top 5 principal components (PCs) of the feature space consisting of the 14 metrics derived from the real cell images for all the cells in the test set. c) Loading of each of the 14 features in the top 5 PCs of the feature space. d) $R^2$ (explained variance) scores of linear regression models that fit the important latent space dimensions to each of the the top 5 PCs, independently, as a function of $\beta$. The black dotted line represents the total explained variance. Since the first five PCs capture 93% of the variation among the 14 features, the theoretical maximum of the black dotted line is 0.93.
(TIF)

## Acknowledgments

We thank Rick Horwitz for scientific discussions. We thank Neda Bagheri and Irina Mueller for careful reading and commenting on the manuscript. We thank Graham Johnson and Thao Do for help with visualization. We thank Ritvik Vasan for help with the Jupyter notebooks. We thank Roy Wollman for valuable advice. We thank the Allen Institute for Cell Science team for creation of the data that enabled this study and for helpful scientific discussions and support. Researchers from the Allen Institute for Cell Science wish to thank the Allen Institute founder, P. G. Allen, for his vision, encouragement and support.

## Author Contributions

**Conceptualization:** Rory M. Donovan-Maiye, Julie A. Theriot, Mary M. Maleckar, Theo A. Knijnenburg, Gregory R. Johnson.

**Data curation:** Rory M. Donovan-Maiye, Jackson M. Brown, Liya Ding, Calysta Yan, Gregory R. Johnson.

**Formal analysis:** Rory M. Donovan-Maiye, Jackson M. Brown, Caleb K. Chan, Liya Ding, Calysta Yan, Gregory R. Johnson.

**Methodology:** Rory M. Donovan-Maiye, Gregory R. Johnson.

**Project administration:** Mary M. Maleckar, Theo A. Knijnenburg.

**Resources:** Nathalie Gaudreault.

**Software:** Rory M. Donovan-Maiye, Jackson M. Brown, Gregory R. Johnson.

**Supervision:** Julie A. Theriot, Mary M. Maleckar, Theo A. Knijnenburg.

**Visualization:** Rory M. Donovan-Maiye, Caleb K. Chan, Julie A. Theriot, Theo A. Knijnenburg, Gregory R. Johnson.

**Writing – original draft:** Rory M. Donovan-Maiye, Mary M. Maleckar, Theo A. Knijnenburg, Gregory R. Johnson.

**Writing – review & editing:** Rory M. Donovan-Maiye, Caleb K. Chan, Nathalie Gaudreault, Julie A. Theriot, Theo A. Knijnenburg, Gregory R. Johnson.

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
