## [Decision Letter · Decision Letter 0]

22 Jun 2021

Dear Dr. Knijnenburg,

Thank you very much for submitting your manuscript "A deep generative model of 3D single-cell organization" for consideration at PLOS Computational Biology. As with all papers reviewed by the journal, your manuscript was reviewed by members of the editorial board and by several independent reviewers. The reviewers appreciated the attention to an important topic. Based on the reviews, we are likely to accept this manuscript for publication, providing that you modify the manuscript according to the review recommendations.

There was enthusiasm from all three reviewers. They also all had suggestions that would improve the manuscript. Most importantly, the authors should further describe the biological significance of the generated models. There was interest in the predictive coupling of subcellular structures, but this should be better described in terms of impact for cell biology. Likewise, biological interpretation of the drug studies would strengthen the paper.

Sincerely,

Jeffrey J. Saucerman

Associate Editor

PLOS Computational Biology

Jason Haugh

Deputy Editor

PLOS Computational Biology

[LINK]

Reviewer's Responses to Questions

**Comments to the Authors:**

Reviewer #1: Overall impressive paper that builds on previous work from this group to learn a 3D generative model from ~10^4 3D images. To my knowledge this is the largest automated analysis of 3D images yet reported. I am impressed by the scale of the dataset/analysis, the technology applied, and the biological realism of the model learned. However, at the end I am left wondering what exactly is the point of the exercise. I am not sure what can be done with this model that couldn’t be done before.

1. the stated practical applications of the research are representation learning and dimensionality reduction. However, nearly the entire paper is devoted to showing the model can learn to generate cell data distributions and that the correlations between the substructures are biologically realistic. But as far as I understand it, the goal of the method is *not* to generate realistic looking cell images (see point 4), but rather to be used for feature representations and dimensionality reduction. However, as far as I can tell, the only evidence presented that these goals can be achieved with this method is in figure 6. But this section is introduced as a demonstration of the use “out of sample data” (which is commendable), which leaves the reader still wondering whether/how well does the generative model work for the original goals of representation learning and dimensionality reduction (even on the held out “in sample” data). No comparisons with other feature representation or dimensionality reduction methods are presented. Therefore, in my mind, the paper does not provide convincing evidence for the stated aims.

2. The introduction does not provide adequate context for the work.

a) What is the current state of the art for 3D images? The authors don’t even distinguish between the methods applied to 3D and 2D images in the introduction.

b) what are other applications of generative models in cell image analysis? In the introduction of previous work all deep learning work is lumped together:

“learning representations of the localization of many 80

independently labeled subcellular structures. This allows us to combine experiments of 81

individual subcellular structures to predict distributions of fluorescent labels that are 82

not directly observed together, creating a single model of integrated cell organization. 83

This approach is distinct from other methods described above, as it can be used to learn 84

and measure population distributions of cellular geometries and organelle localizations 85

within cells, and explore their relationships to one another, as compared to prediction of 86

an expected localization pattern in a given microscopy image.”

It’s not clear which methods “above” the authors are referring to. The clarity of the introduction could be improved if the authors distinguished between generative models on raw images, versus generative models built on feature spaces. Many of the limitations that the authors discuss are more specific to the latter. While the authors focus on more end-to-end and classification applications of deep learning models, there is a lot of recent work that focuses more on image generation (some of the more classic approaches e.g. Osokin et al. 2017 are cited in their discussion, but some artificial fluorescent labeling applications come to mind as well like Christiansen et al. 2018 or Ounkomoi et al. 2018). Additionally, works focus on the relationship between generative modeling and the utility of the latent space for downstream analyses (e.g. Goldsborough et al. 2017) and some specifically interrogate this in a conditional set-up, either exploiting protein-to-cell-structure relationship similar to this paper (Lu et al. 2019) or a drug-to-morphology relationship (Dai Yang et al. 2020). I'm not exactly sure what the authors mean when they say that their work complements previous deep-learning efforts - is it that their model is more statistically principled, more controllable, in a different application space (e.g. 3D images), etc? More specificity on this would be helpful.

3. While the authors show how varying the latent space of their reference structure model changes the morphology of the cell in an interpretable and constrained way, I still think it would be more useful to the reader to know how varying the latent space of the target structure affects the generated images – it seems like a major application of this work is to impute localization structures into existing images of reference cells, after all. Supporting this application, I would be interested in:

a) What kinds of variation are controlled by the latent variables in the target structure variation? Additionally, I would like to see some exploration or discussion of how this relates to the diversity of the training data. I see that the structures consist of training data from a single representative gene, so I expect that the variation will correspond to variation most commonly observed in the genes. For example, the expression level may be relatively constant within a marker versus if there were multiple markers of varying expression, so the range of intensity levels expressed by the generative model may be relatively constrained. Is this a limitation of the model? If so, it should be discussed.

b) Importantly, is the variation described by the target structure variation vector z_t shared across all targets t (e.g. would changing a variable interpreted as cell height for one structure change cell height across all structures), or is the model using factors independently depending on upon cell structure? Knowing this would help interpretation, because I would be able to know if I have to independently interpret the variables for each individual structure, or if just interpreting the variables for one structure is enough to characterize the behavior of the model across all structures.

4. a notable difference between this work and the previous work from these authors is the move away from adversarial losses (and therefore less realistic generation of images). This needs to be discussed.

Minor points

1. The authors devote most of the paper to demonstrating that the model really can learn to generate 3D cell data. Further, figure 3 shows convincingly that the trade off between model complexity and realistic image generation appears as expected in their model as a function of the penalty. However, we have no baseline or expectation for how realistic these images need to be (and for what application). Hence, I would suggest that the authors reduce the emphasis on the generation of “realistic” images.

2.The authors investigate how their parameterization of beta induces a trade-off between the sparsity of the latent space versus the reconstruction quality of the generated images. However, one additional possibility is that the increased penalty from higher parameterizations of beta could be disincentivizing the model from learning more subtle axes of (still biologically important) variation (so while I agree that fewer features are more interpretable, there's a risk that fewer features means the model might miss important aspects of morphology.) I would be interested in seeing the correlation analysis with previous hand-crafted features presented in Fig S3A repeated over the models with different parameterizations in Section 4.3. Are there any kinds of hand-crafted features that are systematically present (e.g. correlated with) the features learned by models with lower parameterizations of beta, that aren't in models with higher parameterizations?

3. The authors try to analyse what they refer to as “out of sample” data, which is commendable. However they are not clear what they mean by out of sample in this context. As far as I can tell, the new dataset was collected under the same conditions, same microscope, same markers., etc. The authors should clarify what they mean by “out of sample”. It seems like a major limitation of the model is the requirement of explicit declaration of the target type, and that the model cannot generalize to target types unseen in training data. For example, if I wanted to generate a different target structure not present in the training data, or if I wanted to generate a multi-localizing protein instead of a discrete localization, this does not seem to be possible in the author's set-up. Relating to this point, it would be very nice to have some discussion/analysis of whether the approach as implemented could be applied to different types of 3D image data from other labs, etc.

Reviewer #2: This is an impressive study that builds on the prior work of the Allen Institute for Cell Science. It learn improved generative models of subcellular patterns. Perhaps the most exciting aspect is the analysis of the dependence of different patterns on the cell and nuclear geometries. The authors also demonstrate that the learned model can be applied to measure changes in drug-treated cells without retraining. The manuscript would be improved by addressing more directly the question of the quality of the generated patterns.

Major points

A major issue in constructing generative models is how to assess how similar generated patterns are to the patterns used to train the model. While reconstruction error can give some indication of this, it does not help with evaluating new synthetic patterns. The manuscript argues that the generated patterns are “biologically plausible” but this is a very weak criterion. In the earliest work on building conditional generative models of subcellular patterns (https://doi.org/10.1002/cyto.a.20487), numerical “SLF” features that had been demonstrated to be able to distinguish all major patterns were used to determine how distinguishable the synthetic images are from the original images. These were also used as a comparison standard in the Human Protein Atlas’ Project Discovery (https://doi.org/10.1038/nbt.4225). There are open source implementations of these features in matlab (https://github.com/CellProfiling/FeatureExtraction) and python (https://mahotas.readthedocs.io/en/latest/features.html). This issue is especially important since for some of the patterns used in this study (and previous work from the Allen Institute) reasonable synthetic patterns are notoriously hard to generate.

Another suggestion approach to improving evaluation is to measure the overlap between the patterns generated for different structures from the same cell/nuclear geometry. Ideally, the synthetic patterns would be somewhat distinct – patterns that are being faithfully generated should show extensive overlap among randomly generated synthetic images of the same structure and lower overlap between different structures.

Minor points

The supplementary figure numbering should be adjusted so that they are numbered in the order that they are referred to in the manuscript.

The introduction says that the work enables generation of a statistically meaningful “average” cell but this is not demonstrated in the manuscript and would be on very shaky ground since the models being learned assume independence of the patterns of the different subcellular structures and this is certainly not correct.

Robert F. Murphy

Professor of Computational Biology Emeritus

Carnegie Mellon University

Reviewer #3: Summary:

--------

The paper presents a method to learn a generative model of cell morphology of different subcellular structures from microscopy image data and to use such models to investigate biologically informative spatial correlations between these. Specifically, the authors first train a variant of an variational autoencoder (beta-VAE) on 3D images of cells labeled with a nuclei and plasma membrane reference marker that then i) is used to generate biologically plausible images by sampling the latent space, and ii) whose latent space is shown to correspond to interpretable cell features (such as height or cell cycle). Next, a similar VAE is trained for each of 24 subcellular markers this time conditioned on the reference marker channels, which again is shown to i) generate realistic images and ii) can be used to quantify the predictive coupling between the subcellular marker and each of its two corresponding reference channels. Finally the generalizability of the learned generative model is shown by applying it on images of a drug perturbation experiment.

Overall:

--------

The paper is generally very well written and I enjoyed reading it.

In my opinion it is an important question of i) how to generate a data-driven statistical model of cell morphology based on microscopy images and ii) how such a data driven model can be used to actually infer biologically meaningful correlations between different substructures, and I think the authors succeeded to convincingly demonstrate that both can be achieved. The presented method based on (standard) beta-VAE is sound and the experiments that demonstrate the usefulness are well carried out and convincing. I especially liked the idea of using the generative model to quantify the "predictive coupling" of the subcellular structures and the reference channels.

Overall, I don't have any major issue with the paper and I think it is a valuable contribution to the journal - please see below my minor comments that I think should be easy to address.

Minor Comments:

---------------

- In Fig2a and b, I would like to see some real example image from the training dataset for some of the depicted groups (e.g. interphase/anaphase for Fig2a and small/large cell height in Fig2b). This would be helpful in comparing with the variations generated from the latent space in Fig2c/d.

- In Fig2c, the feature mu_71 is meant to encode integrated DNA intensity, yet the images generated with increasing mu_71 are showing the DNA channel to get more compact (which is expected) but not to increase in in intensity (which I would have expected, due to chromatin condensation). Is this due to normalization of the shown images? Could the authors comment on this?

- For the perturbation experiment shown in Fig 6, I understand that the cell images in Fig 6c are images sampled from the model, correct? I think it would be helpful to additionally show at least one example images for each condition (Brefeldin and Paclitaxel) from the actual used dataset.

- Was there any specific reason to choose 512 as the latent dimension? The authors write in L152 that there are different "effective latent dimensions". What is meant by "effective"? From FigS6 it appears that beyond > 128, there is only minor improvement (at least for the KL divergence).

- The number of parameters of the used models should be stated in section 6.2

- The overall training time for the full 3D model is (as acknowledged by the authors) quite long (2 weeks). What stopping criterion was used? Maybe the authors could add another sentence discussing this.

- typo in Fig2: "but not showing latent space..." -> "but now showing latent space..." ?

**Have the authors made all data and (if applicable) computational code underlying the findings in their manuscript fully available?**

Reviewer #1: None

Reviewer #2: Yes

Reviewer #3: Yes

PLOS authors have the option to publish the peer review history of their article (what does this mean?). If published, this will include your full peer review and any attached files.

Reviewer #1: No

Reviewer #2: **Yes: **Robert F. Murphy

Reviewer #3: No

Figure Files:

Data Requirements:

Reproducibility:

References:

---

## [Decision Letter · Decision Letter 1]

26 Oct 2021

Dear Dr. Knijnenburg,

Thank you very much for submitting your manuscript "A deep generative model of 3D single-cell organization" for consideration at PLOS Computational Biology. As with all papers reviewed by the journal, your manuscript was reviewed by members of the editorial board and by several independent reviewers. The reviewers appreciated the attention to an important topic. Based on the reviews, we are likely to accept this manuscript for publication, providing that you modify the manuscript according to the review recommendations.

Specifically, the authors should appropriately address the two remaining comments from Reviewer 1, which appear to require text revisions.

Sincerely,

Jeffrey J. Saucerman

Associate Editor

PLOS Computational Biology

Jason Haugh

Deputy Editor

PLOS Computational Biology

[LINK]

Reviewer's Responses to Questions

**Comments to the Authors:**

Reviewer #1: The authors have greatly improved the manuscript. They have clarified that the model cannot (and is not meant) to produce realistic cell images, and have now clarified how their work fits into the context of other recent work in the field. However, there are still 2 areas of the manuscript that I believe need to be improved because they give a misleading impression to the reader.

1. In the abstract, the authors claim: "Once trained, our model can be used to

impute structures in cells where they were not imaged and to quantify the variation in

the location of all subcellular structures..."

I believe the use of the word "impute" is very misleading here. Imputation is used in statistics to refer to the inference of actual values of variables that were not observed. Several recent studies, most famously Christiansen et al. (ref 9) have actually tried to add the labels to images that were not actually observed (could be called true imputation). However, in this study, no evidence is presented that the model can "impute" the locations of structures that were not observed. To do this, the authors would need to hold out the structure channels, and compare the locations of the observed labels to the distribution of labels in the generated images. The metric would simply be R2 or some other reconstruction accuracy. As discussed by the authors, the accuracy would be expected to be very high for the nuclear envelope (because of the tight coupling with the DNA stain) and very low for things like mitochondria and golgi. If they actually did this analysis, it would be clear that the claim in the abstract is misleading: only "tightly coupled" structures can actually be imputed, and the claim in the abstract could be qualified to appropriately reflect the number of structures that can be imputed.

If the authors do not wish to perform a reconstruction analysis and actually report the power of the model to impute the different structures, they at least have to remove or modify this claim in the abstract. Perhaps they can say that their model can predict the statistics of cellular structures that were not imaged, or predict plausible locations where structures might be. But as it stands, the claim of "imputation" is very misleading.

2. The authors have now clarified that the drug treatments they present is an interesting use case of their model: can they detect the effects of the drugs on the structures the drugs are known to perturb. They show convincingly that the latent space of the model is able to detect the changes to the affected compartments.

[Note that there is no "baseline" presented for this analysis so that the reader can grasp the difficulty or potential utility of this feat. The input data for this experiment includes the labeled channels for some structures. Hence, a naive baseline approach here would be to encode maximum intensity projections of the single cell images with ImageNet features (or even classical image features) and look for changes in (say a 2D representation) of the latent space. How hard would it be to see the effects of the drug treated cells on the expected target structure? Regardless of the baseline chosen, the reader needs some context to interpret these results. Could any latent space detect these changes?]

More importantly, the authors write in this section:

"Specifically, the microbule latent space embeddings for the 348

paclitaxel-treated cells show a significant shift in the latent space positions of the overall 349

population, such that the centroid of the population of drug-treated cells is far removed 350

from the latent space origin (Fig. 6d)."

It is still not clear from the manuscript whether the microtubule labels were used in this analysis. Later, the authors write:

"Importantly, the results of this pilot experiment 363

suggest that the model is capable of producing reasonable latent space embeddings for 364

structures that are outside of the range of the original training set (specifically the 365

microtubules for paclitaxel-treated cells and the Golgi for brefeldin-treated cells). 366"

Does this mean that the changes in the microtubules were detected *without* microtubule labeling? If so, this is a great result, and needs to be clarified and highlighted. Although I may be confused by the wording, I don't think is what they have done. I think that the authors are referring to their experiment where they showed that the model correctly predicted no effect on the golgi for the drug that is supposed to affect microtubules. At the very least, the authors need to clarify what was done. Predicting no effect is not nearly as convincing a result as correctly predicting an effect.

I would strongly encourage the authors to hold out the microtubule data from paclitaxel-treated cells, and then see if they can predict (statistically) the effects of the drug on microtubules (using the latent space inferred from tight junctions and golgi.) This could be compared with the statistics of the microtubule staining (which they have). This type of analysis would actually demonstrate that their model can do something that ImageNet features really can't. As it stands, although I am convinced that the authors have built a beautiful model of the cell using unprecedented data, they have still not really demonstrated that this model can provide any new insight or practical utility.

Reviewer #2: The manuscript has been revised to address the reviewer concerns.

Reviewer #3: In the revised manuscript the authors have addressed all my issues.

**Have the authors made all data and (if applicable) computational code underlying the findings in their manuscript fully available?**

Reviewer #1: Yes

Reviewer #2: Yes

Reviewer #3: Yes

PLOS authors have the option to publish the peer review history of their article (what does this mean?). If published, this will include your full peer review and any attached files.

Reviewer #1: No

Reviewer #2: No

Reviewer #3: No

Figure Files:

Data Requirements:

Reproducibility:

References:

---

## [Editor Report · Decision Letter 2]

29 Nov 2021

Dear Dr. Knijnenburg,

We are pleased to inform you that your manuscript 'A deep generative model of 3D single-cell organization' has been provisionally accepted for publication in PLOS Computational Biology.

Best regards,

Jeffrey J. Saucerman

Associate Editor

PLOS Computational Biology

Jason Haugh

Deputy Editor

PLOS Computational Biology

---

## [Editor Report · Acceptance letter]

3 Jan 2022

PCOMPBIOL-D-21-00930R2 

A deep generative model of 3D single-cell organization

Dear Dr Knijnenburg,

I am pleased to inform you that your manuscript has been formally accepted for publication in PLOS Computational Biology. Your manuscript is now with our production department and you will be notified of the publication date in due course.

With kind regards,

Livia Horvath
